# Association between Weather and Self-Monitored Steps in Individuals with Prediabetes and Type 2 Diabetes in Sweden over Two Years

**DOI:** 10.3390/ijerph21040379

**Published:** 2024-03-22

**Authors:** Yohannes Woldamanuel, Patrick Bergman, Philip von Rosen, Unn-Britt Johansson, Maria Hagströmer, Jenny Rossen

**Affiliations:** 1Department of Health Promoting Science, Sophiahemmet University, 114 86 Stockholm, Sweden; unn-britt.johansson@shh.se (U.-B.J.); maria.hagstromer@ki.se (M.H.); jenny.rossen@shh.se (J.R.); 2Department of Medicine and Optometry, eHealth Institute, Linnaeus University, 391 82 Kalmar, Sweden; patrick.bergman@lnu.se; 3Department of Neurobiology, Care Sciences and Society, Karolinska Institutet, 141 83 Stockholm, Sweden; philip.von.rosen@ki.se; 4Academic Primary Care Center, Region Stockholm, 113 65 Stockholm, Sweden

**Keywords:** physical activity, precipitation, prediabetes, steps, sunshine, temperature, type 2 diabetes

## Abstract

Background: Many studies have identified key factors affecting the rates of engagement in physical activity in older adults with chronic disease. Environmental conditions, such as weather variations, can present challenges for individuals with chronic diseases, such as type 2 diabetes when engaging in physical activity. However, few studies have investigated the influence of weather on daily steps in people with chronic diseases, especially those with prediabetes and type 2 diabetes. Objective: This study investigated the association between weather variations and daily self-monitored step counts over two years among individuals with prediabetes and type 2 diabetes in Sweden. Methods: The study is a secondary analysis using data from the Sophia Step Study, aimed at promoting physical activity among people with prediabetes and type 2 diabetes, which recruited participants from two urban primary care centers in Stockholm and one rural primary care center in southern Sweden over eight rounds. This study measured physical activity using step counters (Yamax Digiwalker SW200) and collected self-reported daily steps. Environmental factors such as daily average temperature, precipitation, and hours of sunshine were obtained from the Swedish Meteorological and Hydrological Institute. A robust linear mixed-effects model was applied as the analysis method. Results: There was no association found between weather variations and the number of steps taken on a daily basis. The analysis indicated that only 10% of the variation in daily steps could be explained by the average temperature, precipitation, and sunshine hours after controlling for age, gender, and BMI. Conversely, individual factors explained approximately 38% of the variation in the observations. Conclusion: This study revealed that there was no association between weather conditions and the number of daily steps reported by individuals with prediabetes and type 2 diabetes taking part in a physical activity intervention over two years. Despite the weather conditions, women and younger people reported more steps than their male and older counterparts.

## 1. Introduction

Research strongly suggests that physical activity promotes healthy aging [1,2]. According to a 2016 estimation, approximately 28% of the world’s adult population is insufficiently physically active [3]. Individuals with type 2 diabetes are typically less physically active than those without this condition [4,5], and this can negatively affect their health. The evidence indicates that regular physical activity can effectively manage blood glucose levels and delay and even prevent complications in individuals with prediabetes and type 2 diabetes [6]. It is crucial to comprehensively understand the factors affecting these groups’ physical activity levels to develop effective strategies for increasing engagement in physical activity and promoting healthy aging.

The key factors associated with physical activity engagement in older adults are demographics, personal and behavioral characteristics, psychological and social aspects, and physical environmental factors [7,8,9]. Physical environmental factors play an important role in shaping older adults’ physical activity behaviors [10]. Physical environmental factors include built environments like road safety and walking facilities, residential neighborhoods [9,11], as well as natural environmental factors such as weather, terrain, and vegetation [12,13]. The main challenges of promoting physical activity in older adults are posed by uncontrollable environmental conditions such as weather and day length [14,15]. In addition, physical decline and health issues are common during aging, and these phenomena are sensitive to adverse weather conditions, which could negatively affect older adults’ physical activity levels [15,16].

Adverse weather conditions are associated with a higher mortality and morbidity risk rate in individuals with chronic health conditions [17]. This negative correlation has been observed in outcomes related to cardiovascular issues and other health concerns linked to high- or low-temperature weather conditions [18,19]. Therefore, gathering evidence on how weather affects physical activity levels is imperative, particularly for those with chronic health conditions, as the information can help individuals make informed decisions about their outdoor activities and enable healthcare professionals to provide more favorable and affordable prescriptions for outdoor exercise and as it is crucial for individuals with chronic health conditions to reach the recommended level of physical activity [20]. Various studies have investigated the correlation between weather conditions and physical activity behavior. Some studies have utilized self-reports to evaluate physical activity levels [18,21,22], while others have objectively assessed daily physical activity over a short period [23,24,25]. Therefore, the need for long-term, objectively measured physical activity assessments with different weather conditions is paramount to overcome the limitations created by self-reported bias.

Although previous weather-related studies have not specifically focused on populations with chronic diseases, many have examined how weather and seasonal changes impact objectively measured physical activity levels in older adults [21,26,27]. For instance, Klenk et al. found a strong positive association between the daily maximum temperature and global radiation and objectively measured walking among older adults, but found no strong correlation with daylight length [28]. A similar study conducted with individuals aged 65 and over showed that they tend to spend more time outside and walk for longer on warm, sunny days [29]. Although weather data were collected daily and physical activity was measured objectively in these studies, they were short-term studies.

To our knowledge, few studies have investigated the influence of weather conditions on daily steps in people with prediabetes or type 2 diabetes. A study by Dasgupta et al. explored the impact of seasonal changes on walking patterns and vascular risk factors in people with type 2 diabetes. The results showed that during the fall and winter months, daily steps decreased. Furthermore, this study found a negative correlation between daily steps and systolic blood pressure but showed no significant seasonal differences concerning Hemoglobin A1c levels [30].

The current study is a secondary analysis of the Sophia Step Study. The primary study revealed no intervention effect on cardiometabolic risk factors, although the intervention maintained moderate to vigorous physical activity over two years. However, there was significant variation in individual activity levels [31].

This study investigates the association of weather variations with daily self-monitored steps over two years among individuals with prediabetes and type 2 diabetes in Sweden.

## 2. Materials and Methods

This study is a secondary analysis using data from the two intervention groups participating in the Sophia Step Study, a randomized controlled trial (RCT) aimed at promoting physical activity among people with prediabetes and type 2 diabetes. Participants were recruited from two urban primary healthcare centers in Stockholm and one rural primary healthcare center in southern Sweden over eight rounds. The first-round participants began the study in February 2013, while the last group started in January 2018. Those interested in participating were given information about the study and were booked for baseline assessment. A total of 385 participants were invited, and a general practitioner assessed each participant’s eligibility. A study protocol providing a detailed description of the Sophia Step Study has been published elsewhere [32].

**Inclusion criteria:** HbA1c > 39 mmol/mol or fasting blood glucose > 5.6 mmol/L, duration of type 2 diabetes diagnosis > 1 year, age 40–80, and ability to read, write, and express themselves in the Swedish language.

**Exclusion criteria:** myocardial infarction in the past six months, serum creatinine > 140, diabetic ulcer or risk for ulcers, prescribed insulin in the past six months, additional disease prohibiting physical activity, repeated hypoglycemia or severe hypoglycemia in the last 12 months, classified in the very-high-intensity activity group according to Stanford Brief Activity Survey [33], and no access to the internet.

In total, 203 participants were included as study participants before randomization. After applying the exclusion criteria, the project allocated 188 participants into three arms: 64 were assigned to the multiple-component group, 59 to a single-component group, and 65 to a standard care group. The multi-component group received daily step-monitoring support and individual and group counseling sessions, while the single intervention group only received a daily step-monitoring device. The standard care group did not receive a monitoring device. As a result, the current study only analyzed the daily step data of the two intervention groups, which included 120 participants.

Participants in the intervention groups were given step counters (Yamax Digwalker SW200: manufactured by Yamax Corporation Tokyo, Japan) and asked to monitor their daily steps using a website [34]. The participants were instructed to log their daily steps in a paper diary and then submit their logs weekly to the website for two years. In case any issues were encountered with the step counters or if it was lost, a new one could be obtained from the healthcare center or sent via post.

As the individuals reported their own steps, we found some typographical errors that required correction. As a result, we examined the data for outliers and concluded that any values below 200 or above 40,000 steps per day were unrealistic and treated them as missing data.

Weather data were obtained from the Swedish Metrological and Hydrological Institute. The data included the daily temperature in degrees Celsius (°C), daily precipitation in millimeters (mm), and daily sunshine (in seconds converted to hours). The study focused on two geographical regions, Stockholm and Kalmar, and all metrological stations were within a 20 km radius of the participants’ residences. A questionnaire with items on gender, diet, educational levels, and income collected demographic data at baseline. Weight and height were measured wearing light clothes at baseline.

Given multiple observations within each individual, a linear mixed model is an appropriate analysis, i.e., the data are not independent such as the case is in longitudinal studies [35]. However, due to the presence of several influential observations which may distort the relationships when using an ordinary linear mixed model, the data were modeled using a robust linear mixed model approach. Robust linear models can be used to remove some of the effects in such cases by weighting the data, giving influential cases less importance [36]. We modeled the association between steps (the dependent variable) and daily temperature, precipitation, and hours of daily sunshine (independent variables). The model was further adjusted for sex, age, intervention group, and body mass index (BMI). Due to the fact that the residuals were not normally distributed, a Box–Cox transformation was conducted, which indicated that a square root transformation of the dependent variable was suitable to satisfy the assumption of normally distributed residuals [37]. The *p*-values from the robust linear mixed model were calculated using the Satterthwaite method [38]. Figures were made using the ggplot2 package in R [39].

## 3. Results

The daily step data from 120 participants collected over two years were analyzed. Table 1 shows that 58% (*n* = 70) of the participants were men, the average age was 64.6 (sd. = 7) years, and 82% (*n* = 98) had type 2 diabetes, with an average disease duration of 8.7 (sd. = 6.3) years at baseline. The participants had a mean BMI of 29 (sd. = 4.3) kg/m^2^ and a mean waist circumference of 103 (sd. = 13.3) cm at baseline.

The mean daily step count at baseline was 7441 (sd. = 4157). Figure 1 shows the mean daily steps per intervention group during the Sophia Step Study intervention period. Over the two years, the mean temperature was 8.6 (sd. = 7.2) (°C), the mean precipitation per day was 1.5 (sd. = 3.8) mm, and the average sunshine hours in one day were 5.5 (sd. = 5).

Figure 2 illustrates the association between step counts and weather variables, considering the participants’ age (below or above 65), gender (men or women), and body mass index (BMI) for normal weight, overweight, and obese individuals. Participants under the age of 65 had slightly higher daily step counts than those over 65. Moreover, women had slightly higher step counts as compared to men. These differences were not significant but remained consistent across various weather conditions such as temperature, precipitation, and sunshine duration. Finally, there was no association between weather conditions and step counts across different BMI levels.

Table 2 presents the results of the robust linear mixed-effect model, showing that the independent variables (temperature, precipitation, and sunshine) accounted for 10% of the variation in daily steps after controlling for age, gender, and BMI. Conversely, individual factors explained approximately 38% of the variations in the daily steps. The analysis revealed no relationship between variations in weather conditions and daily step counts. Additionally, the overall beta estimates of the weather variables remained consistently below ±0.3. The intervention group was non-significant (*p* = 0.810) when tested in the model and was not included in the final model.

## 4. Discussion

We examined the influence of different weather conditions on daily steps taken by people with prediabetes and type 2 diabetes during a physical activity intervention. The study did not explain the variations in step counts among individuals with prediabetes and type 2 diabetes during the two-year physical activity intervention based on weather conditions. However, the analysis revealed a significant relationship between weather variables and daily steps, mainly due to the high number of observations being analyzed (around 740 observations per individual). Additionally, the study found that higher temperatures, longer sunshine hours, and lower precipitation rates tended to increase daily step counts insignificantly. On the other hand, when weather conditions were unfavorable, there was a decreasing trend in daily steps.

The current study found that participants under the age of 65 tended to record more daily steps than those over 65. Women appeared to have a slightly higher and more consistent step count than men irrespective of weather conditions, but the difference was insignificant. The study also indicated that these differences in step counts remained consistent across various weather conditions, such as temperature, precipitation, and sunlight duration. While previous studies have suggested that men tend to walk more as a leisure activity than women [40,41], this study presents opposing findings that challenge this notion. There was no significant difference in step counts between the intervention groups in our current study, which is consistent with the previous results of the two intervention groups of the same RCT [31]. In addition, we found no association between weather conditions and step counts across different BMI levels and intervention groups in our study population.

A study investigating how weather affected the physical activity levels of older adults found that warmer temperatures and lower rainfall increase walking among seniors [42]. Similarly, another study conducted in Norway used accelerometers and hourly weather data to measure physical activity levels and discovered that increased temperatures have a positive impact on physical activity, while higher precipitation has a negative effect [16]. It should be noted that the previous studies only collected objective physical activity data for short periods, while this current study considered two years of self-reported steps. The discrepancies in physical activity outcomes may be due to differences in measurement methods, data collection types, period lengths, and weather conditions considered.

A two-year phone-based survey on physical activity found that participants were less likely to engage in outdoor physical activity in extreme hot and cold temperatures [19], which contradicts the current study’s findings without considering extreme weather conditions. However, this study also found that severe cold weather did not have an impact on the activity level of individuals with non-communicable diseases (NCDs), including type 2 diabetes, except for those with cardiovascular disease [19]. To understand the health implications for those with chronic illnesses, long-term studies that thoroughly examine the relationship between extreme weather and physical activity may be necessary.

It appears that physical activity levels are influenced by various factors rather than solely by weather variations. The results of the current study indicate that the amount of precipitation during the intervention was not significantly correlated with the number of steps taken. A previous study that analyzed the relationship between physical activity behavior and weather conditions using a wrist-worn Fitbit activity tracker over 13 months discovered that physical activity intensity played a crucial role in engaging in activity in adverse weather conditions. For instance, precipitation affected adults who partook in moderate to vigorous physical activity significantly more than those who partook in light physical activity [43]. This may be because weather conditions such as ice, snow, or severe cold would be more of an impediment for higher-intensity activities like running than for lower-intensity activities like walking. In addition, a study conducted by Aspvik et al. suggested that physically fit individuals may not be as affected by precipitation and other weather conditions as physically unfit individuals [16]. In the present study, we did not take into account exercise intensity or fitness levels but found no correlation between weather conditions and BMI.

A recent study aimed to investigate the possible correlation between weather conditions and outdoor walking in older adults with the activPAL sensor over the course of a week. The study found a strong correlation between longer sunny days and the participants’ inclination to spend more time outdoors and walk for longer periods of time [29]. This is in contrast to the findings of the current study, in which a clear association with sunshine was not observed.

In the current study, the participants were involved in a pedometer intervention program that lasted for over two years. The program involved using a pedometer to set a daily step goal and monitor daily steps. An interview study conducted with a sample of the participants revealed that the pedometer was a motivating factor and that they had adopted problem-solving techniques to achieve their goals [44]. The improved accessibility and accuracy of weather forecasting through digital platforms and public broadcasting services and warnings help individuals make informed decisions about when to be active [24,45], which may have contributed to the lower impact of weather on their physical activity. In addition, Ferguson et al. (2023) suggested that light physical activity is not significantly affected by extreme weather if proper clothing and equipment, such as an umbrella, are used to address the challenge [43]. The factors above highlight some of the significance of having effective adaptation strategies to cope with different weather conditions at the individual level [46]. Similarly, the current study’s participants were individuals with prediabetes or type 2 diabetes who were in relatively good health and had rather high levels of physical activity [31]. It is important to mention that they may have developed coping strategies to manage different weather conditions since they were part of a physical activity intervention.

One of the strengths of the current study is that step counts were measured over two years, which allowed the associations of various weather conditions over an extended time frame to be examined. Secondly, our study used daily weather data on temperature, precipitation, and sunshine to reduce the impact of external factors on seasonal patterns and physical activity. This approach is advantageous, as sudden weather changes are unlikely to occur within a day, minimizing changes in activity decisions.

Some of the limitations of the current study must be addressed. The present study is a secondary analysis of an intervention study which induces some bias to the internal validity. The study utilized step counters to measure the number of steps taken by participants. However, these pedometers were not sensitive enough for individuals with a high BMI, and as a result, some participants’ actual numbers of steps may have been underestimated. Additionally, participants were required to report their daily steps on a website, which could have led to social desirability bias. They were also instructed to convert time spent engaging in activities like swimming or cycling to an equivalent number of steps. It would have been better to keep track of whether the step counts were obtained from indoor, outdoor, or non-ambulant activities. Furthermore, the study did not take into account the impact of surface conditions during cold and icy seasons. Additionally, the pedometers did not measure the intensity, duration, or type of exercise that the participants engaged in [47]. Finally, the study participants were selected from only two geographical regions of Sweden, which limits the geographical diversity of the results. Therefore, it is essential to exercise caution when applying the results of this study to a larger population.

Finally, the current study’s findings reveal that individual factors play a more significant role in influencing self-monitored physical activity than weather conditions. However, it is still important to consider weather conditions while promoting physical activity on a larger scale. By implementing effective interventions and considering alternatives for indoor activities in extreme weather conditions, we can increase physical activity levels and enhance the safety of the participants in different weather conditions. This, in turn, can positively impact people’s well-being and public health.

## 5. Conclusions

Our study found no significant association between weather conditions (temperature, precipitation, and sunshine) and self-reported step counts among individuals with prediabetes and type 2 diabetes taking part in a physical activity intervention over two years. Despite the weather conditions, women and younger people reported more steps compared to their male and older counterparts.

## Figures and Tables

**Figure 1 ijerph-21-00379-f001:**
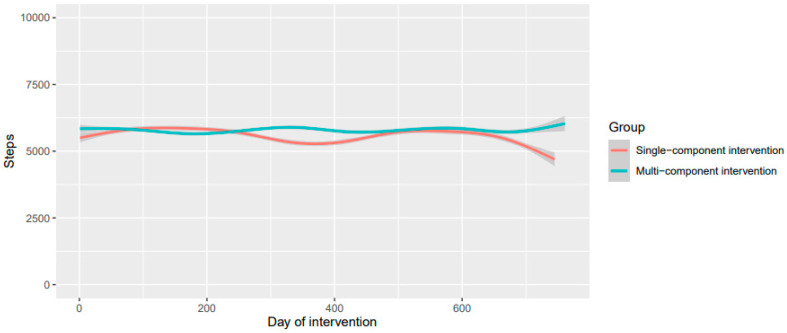
Mean daily steps per intervention group during the Sophia Step Study intervention period.

**Figure 2 ijerph-21-00379-f002:**
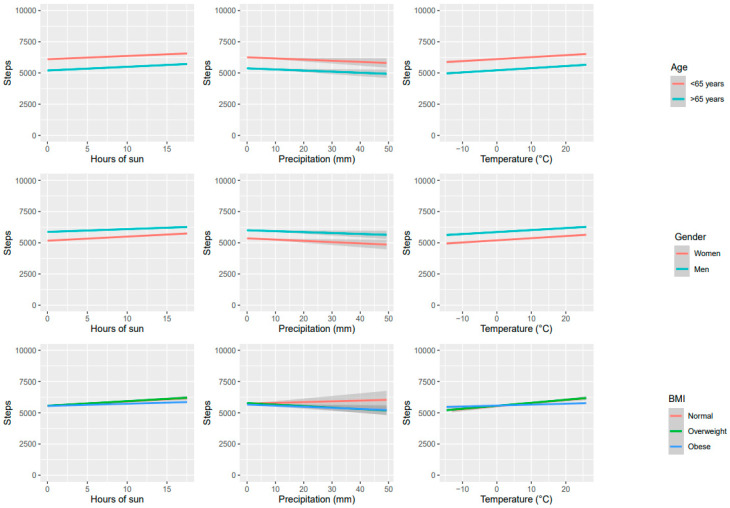
Associations between weather variables and step counts were analyzed by age, sex, and BMI level.

**Table 1 ijerph-21-00379-t001:** Characteristics of participants.

	Tota (*n* = 120)	Women (*n* = 50)	Men (*n* = 70)
Age, years mean (sd)	64.6 (7.0)	64.6 (7.6)	64.6 (6.6)
Prediabetes *n* (%)	22 (18)	15 (30)	7 (10)
Duration of PD, mean (sd)	1.6 (1.3)	-	-
Duration of T2D, mean (sd)	8.7 (6.3)	-	-
Higher education *n* (%)	53 (44.2)	27 (54)	26 (37)
Low income (%)	40 (33.3)	17 (34)	23 (32.8)
BMI, kg/m^2^ mean (sd)	29.1 (4.3)	29.6 (5.5)	30.1 (4.2)
Waist circumference, cm. mean (sd)	103 (12.3)	98.9 (12.4)	107 (11.1)

Presentation of values in mean (sd) or percentage distribution (%). BMI: body max index (kg/m^2^); PD: prediabetes; T2D: type 2 diabetes.

**Table 2 ijerph-21-00379-t002:** Coefficients of associations of weather variables with daily steps, covariates, and 95% confidence intervals from the robust linear mixed model.

	Estimates	SE	95% CI			*p*-Value	T-Value
Precipitation (mm)	−0.06	0.02	−0.10	–	−0.03	<0.001	−3.51
Temperature (°C)	0.08	0.01	0.06	–	0.10	<0.001	6.97
Sunshine (hours)	0.28	0.02	0.25	–	0.31	<0.001	17.25
Age (years)	−0.84	0.19	−1.20	–	−0.47	<0.001	−4.45
Gender (men)	6.33	2.67	1.09	–	11.56	0.018	2.37
BMI (kg/m^2^)	−0.60	0.31	−1.20	–	0.00	0.049	−1.97

## Data Availability

To access the data used in this study, one must follow the data legislation of Sweden and the EU since the datasets are not publicly available. Each request for access will be handled on a case-by-case basis and will require a data transfer and use agreement with the recipient to ensure appropriate regulation.

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
