# Peer review of "Association between Weather and Self-Monitored Steps in Individuals with Prediabetes and Type 2 Diabetes in Sweden over Two Years"

_ijerph, 2024, doi:10.3390/ijerph21040379_

Round 1

Reviewer 1 Report

Comments and Suggestions for Authors

The study reports the relationship between physical activity and weather conditions while the improvement of metabolic factors in patients with pre-diabetes and type 2 diabetes should be evaluated. The statistical analysis is described in a very concise manner, and multiple regression to a reader not versed in statistical methodology may not be easy to understand

1. The authors evaluate the association between weather conditions and
    self-monitored steps in people with prediabetes and type 2 diabetes
    in Sweden over two years by correlating physical activity, measured
    only by the number of daily steps, conditional only on this factor.
    In fact, previous studies and the authors' own study fail to
    highlight this. In fact, physical activity can also be done in
    different forms.
    It's important to note that physical activity can encompass various
    forms beyond just daily steps. Therefore, it might be beneficial for
    the study to consider and incorporate other types of physical
    activity to provide a more comprehensive analysis.
 2. The study also does not evaluate the comparison of metabolic control
    in subjects with variability in the number of daily steps.
    The absence of an evaluation regarding the comparison of metabolic
    control in subjects with varying levels of daily steps represents a
    gap in the analysis. Including such an evaluation could provide
    valuable insights into the relationship between physical activity
    and metabolic health.
3. The multivariate analysis does not examine any of the patients'
    metabolic conditions, blood glucose, glycated hemoglobin, and others.
4. The study does not add to the subject area compared with other
    published studies on the topic.
5. The lack of inclusion of patients' metabolic conditions in the
    multivariate analysis limits the depth of understanding regarding
    the relationship between weather conditions, physical activity, and
    metabolic health. Incorporating these factors into the analysis
    could strengthen the study's findings.
6. The study's failure to contribute new insights or findings to the
    existing body of literature on the topic suggests a missed
    opportunity. Identifying areas where the study diverges from or
    builds upon previous research could enhance its significance and
    relevance.
7. Commenting on the figures and data quality would provide valuable
    feedback on the presentation and reliability of the study's
    findings. Assessing the clarity, accuracy, and completeness of the
    figures, as well as the robustness of the data collection and
    analysis methods, can enhance the overall credibility and impact of
    the study.

Author Response

Response to the review

The study reports the relationship between physical activity and weather conditions while the improvement of metabolic factors in patients with pre-diabetes and type 2 diabetes should be evaluated. The statistical analysis is described in a very concise manner, and multiple regression to a reader not versed in statistical methodology may not be easy to understand.

Author response: Thank you for the comment. We added some clarification on the analysis procedures.

  1. The authors evaluate the association between weather conditions and self-monitored steps in people with prediabetes and type 2 diabetes in Sweden over two years by correlating physical activity, measured only by the number of daily steps, conditional only on this factor.  In fact, previous studies and the authors' own study fail to highlight this. In fact, physical activity can also be done in different forms. It's important to note that physical activity can encompass various forms beyond just daily steps. Therefore, it might be beneficial for the study to consider and incorporate other types of physical activity to provide a more comprehensive analysis.

Author response: Thank you for suggesting the importance of including other forms of physical activity in the study. However, the current study is a secondary analysis of a two-year randomized control trial that only collected data on participants' self-monitored step count data, which was regularly obtained and registered for most days for two years. Therefore, it makes it feasible to explore the relationship with weather conditions. Even we have pointed out some limitations in the method discussion to acknowledge that other forms of physical activity were not accounted for in the study. (refer to lines 277-282)

  1. The study also does not evaluate the comparison of metabolic control in subjects with variability in the number of daily steps. The absence of an evaluation regarding the comparison of metabolic control in subjects with varying levels of daily steps represents a gap in the analysis. Including such an evaluation could provide valuable insights into the relationship between physical activity and metabolic health.

Author response: Thank you for your suggestion. We have already published an article about the effect of the intervention on metabolic health. We refer you to the publication on how the Sophia Step Study (a three-armed randomized controlled trial) intervention affects the participants' metabolic health. The current study focuses solely on examining the association between weather and self-monitored step counts.

Rossen J, Larsson K, Hagströmer M, Yngve A, Brismar K, Ainsworth B, Åberg L, Johansson UB. Effects of a three-armed randomised controlled trial using self-monitoring of daily steps with and without counselling in prediabetes and type 2 diabetes-the Sophia Step Study. Int J Behav Nutr Phys Act. 2021 Sep 8;18(1):121. doi: 10.1186/s12966-021-01193-w.

  1. The multivariate analysis does not examine any of the patients' metabolic conditions, blood glucose, glycated hemoglobin, and others.

Author response: Thank you for the comment, we refer to the response on comment number 2.

  1. The study does not add to the subject area compared with other published studies on the topic.

Author response: We appreciate your comments and suggestions. Based on your recommendations, we have elaborated on the knowledge gap in our study in the introduction by drawing connections to previous research.  Lines 58-72

  1. The lack of inclusion of patients' metabolic conditions in the multivariate analysis limits the depth of understanding regarding the relationship between weather conditions, physical activity, and metabolic health. Incorporating these factors into the analysis could strengthen the study's findings.

Author response: Thank you for the comment. In this study, we decided to focus on the associations between weather and physical activity.

  1. The study's failure to contribute new insights or findings to the existing body of literature on the topic suggests a missed opportunity. Identifying areas where the study diverges from or builds upon previous research could enhance its significance and relevance.

Author response: Thank you for your comment. We appreciate your concern, but we would like to inform you that we have not found any research studies examining the correlation between weather changes and the number of steps taken over a longer period, particularly among our population. However, as you have noted, we have addressed the significance of our study in the research area in lines 58-72.

  1. Commenting on the figures and data quality would provide valuable feedback on the presentation and reliability of the study's findings. Assessing the clarity, accuracy, and completeness of the figures, as well as the robustness of the data collection and analysis methods can enhance the overall credibility and impact of the study.

Author response: Thanks for the suggestion. The manuscript contains tables and a figure explaining the analysis's results. We have commented transparently on the interpretation of the results. For example, the explanation of Table 1 can be found between lines 157-162, while the explanation of Table 2 can be found between lines 179-185. The figure is described in lines 169-176. We also added the method, data collection process, and data analysis in lines 99 - 156.

Reviewer 2 Report

Comments and Suggestions for Authors

In my comments, I mentioned that it was a well-structured manuscript. However, I leave it to the editors to decide whether or not to publish the study, as I do not consider the results relevant to the journal's objectives. Nevertheless, I apologize for not addressing all the specified points. Below are the responses.

In this manuscript the question to be addressed is whether there is an association between climate variations and daily self-monitored step counts over two years among individuals with prediabetes and type 2 diabetes.

In the discussion section, the authors highlight the gap in the literature regarding studies that consider climate variations in outdoor physical activity, but from my standpoint, its relevance to environmental studies is somewhat limited, as it addresses a factor that is not easily modifiable. Although the findings are straightforward, indicating no significant association between weather conditions and the observed trend of women taking more steps than men, I find its significance within the broader context of the journal's focus to be questionable. While the study contributes valuable insights, its implications for advancing our understanding of environmental dynamics may be relatively constrained. Therefore, I defer to the editorial discretion in determining the suitability of this manuscript for publication."

The study offers unique insights into how weather-related barriers may impact physical activity among individuals with prediabetes and type 2 diabetes. The authors report that physical activity levels are influenced by various factors rather than solely by weather variations and it will be challenging to consider other variables such as whether they had comorbidities like depression that might hinder their ability to walk.

While the manuscript effectively examines the relationship between weather conditions and step counts, its focus on factors such as daily temperature, precipitation, and sunshine may not directly align with the core concerns of environmental studies, which often encompass broader issues such as climate change, biodiversity conservation, and sustainability practices. Additionally, the study's findings primarily pertain to individual behavior rather than broader environmental impacts or policy implications.

The conclusions drawn in the manuscript appear to be consistent with the evidence and arguments presented throughout the study. The research rigorously examines the association between weather variations and daily step counts among individuals with prediabetes and type 2 diabetes over a two-year period. The findings suggest that there is no significant correlation between weather conditions and step counts, while also indicating that women tend to take more steps than men, regardless of weather factors. However, it's important to note that while the study addresses these main questions, there may be additional factors influencing physical activity levels, such as comorbidities like depression, which were not explicitly explored in this research. Thus, while the conclusions drawn are supported by the evidence presented, further investigation into the broader contextual factors affecting physical activity could enhance the comprehensiveness of the study's findings.

The references are appropriately selected and accurately cited, providing a comprehensive overview of the existing literature and its relevance to the current study, then they can be considered appropriate. Since 2009 to 2024.

In addition to the consistency of conclusions with the evidence presented, it's worth noting the quality of the tables and figures included in the manuscript. The tables are well-organized and effectively summarize the key data points, facilitating easy comprehension of the results. Similarly, the figures, such as graphs and charts, are visually appealing and clearly illustrate the trends and patterns observed in the data.

Author Response

Respons to Review

In my comments, I mentioned that it was a well-structured manuscript. However, I leave it to the editors to decide whether or not to publish the study, as I do not consider the results relevant to the journal's objectives. Nevertheless, I apologize for not addressing all the specified points. Below are the responses.

In this manuscript the question to be addressed is whether there is an association between climate variations and daily self-monitored step counts over two years among individuals with prediabetes and type 2 diabetes.

In the discussion section, the authors highlight the gap in the literature regarding studies that consider climate variations in outdoor physical activity, but from my standpoint, its relevance to environmental studies is somewhat limited, as it addresses a factor that is not easily modifiable. Although the findings are straightforward, indicating no significant association between weather conditions and the observed trend of women taking more steps than men, I find its significance within the broader context of the journal's focus to be questionable. While the study contributes valuable insights, its implications for advancing our understanding of environmental dynamics may be relatively constrained. Therefore, I defer to the editorial discretion in determining the suitability of this manuscript for publication."

The study offers unique insights into how weather-related barriers may impact physical activity among individuals with prediabetes and type 2 diabetes. The authors report that physical activity levels are influenced by various factors rather than solely by weather variations and it will be challenging to consider other variables such as whether they had comorbidities like depression that might hinder their ability to walk.

While the manuscript effectively examines the relationship between weather conditions and step counts, its focus on factors such as daily temperature, precipitation, and sunshine may not directly align with the core concerns of environmental studies, which often encompass broader issues such as climate change, biodiversity conservation, and sustainability practices. Additionally, the study's findings primarily pertain to individual behavior rather than broader environmental impacts or policy implications.

The conclusions drawn in the manuscript appear to be consistent with the evidence and arguments presented throughout the study. The research rigorously examines the association between weather variations and daily step counts among individuals with prediabetes and type 2 diabetes over a two-year period. The findings suggest that there is no significant correlation between weather conditions and step counts, while also indicating that women tend to take more steps than men, regardless of weather factors. However, it's important to note that while the study addresses these main questions, there may be additional factors influencing physical activity levels, such as comorbidities like depression, which were not explicitly explored in this research. Thus, while the conclusions drawn are supported by the evidence presented, further investigation into the broader contextual factors affecting physical activity could enhance the comprehensiveness of the study's findings.

The references are appropriately selected and accurately cited, providing a comprehensive overview of the existing literature and its relevance to the current study, then they can be considered appropriate. Since 2009 to 2024.

In addition to the consistency of conclusions with the evidence presented, it's worth noting the quality of the tables and figures included in the manuscript. The tables are well-organized and effectively summarize the key data points, facilitating easy comprehension of the results. Similarly, the figures, such as graphs and charts, are visually appealing and clearly illustrate the trends and patterns observed in the data.

Author response: We appreciate your acknowledgment of the well-structured manuscript addressing the literature gap on the topic. However, as mentioned in your comment, we have decided to narrow our research question to how weather affects physical activity behavior. Based on the current study data, it may be challenging to incorporate climate change, biodiversity conservation, and sustainability practices.

The other suggestion was relevant regarding the impact of bidirectionality on physical activity behavior and comorbidities, such as symptoms of depression. However, we did not include it in our analysis as none of the participants had reported depression symptoms throughout the intervention period.

Reviewer 3 Report

Comments and Suggestions for Authors

The submitted paper aimed to investigate the association between weather conditions and daily steps among individuals with prediabetes or type 2 diabetes. The study analysed data from a study that took place in Sweden and participants were assigned randomly to three intervention groups. The main weakness of the submitted article is that the intervention group, to which each participant belongs, was ignored in the analysis carried out and in the discussion of the results obtained.

Lines 81 to 88: Although a reference is included concerning a detailed description of the study, some brief explanation should be provided concerning the recruitment of participants. In the Introduction section authors consider individuals with prediabetes and type 2 diabetes, but in the Material and Methods there is no reference to those conditions.  Authors should also provide some explanation on why 385 participants were invited, but only 120 were included in the analysis. What was the inclusion/exclusion criteria?

Line 89: Specify the intervention groups.

Line 108: Considering the study design, the intervention group should also be considered as an independent variable.

Line 123: The daily step count is the main outcome of this study, but only a descriptive at baseline is given. As the study took two years, I suggest the presentation of a more detailed analysis of the daily step count.

Lines 134 to 136: The intervention group should also be considered in Figure 1.

Lines 144 to 145: In table 2 I suggest the replacement of the T-value by the P-value.

Lines 169 to 171: This statement does not seem to agree with the results presented in Table 2, and the conclusions presented in the first paragraph of the Discussion.

Line 181: The writing does not look right. Please verify.

Lines 189 to 205: Since "The current study is a secondary analysis of the Sophia Step Study, a randomized controlled trial (RCT) aimed at promoting physical activity among people with prediabetes and type 2 diabetes", some discussion should be included regarding the impact of the interventions in daily steps during the two year program. In the evaluation of the impact of weather conditions in the daily steps, the interventions that each participant was subjected should be considered and discussed.

Lines 227 to 232: The conclusions presented in the Conclusions section are not in line with the conclusions presented in the first paragraph of the Discussion section. Please check the text.

Author Response

Respons to Review

The submitted paper aimed to investigate the association between weather conditions and daily steps among individuals with prediabetes or type 2 diabetes. The study analysed data from a study that took place in Sweden and participants were assigned randomly to three intervention groups. The main weakness of the submitted article is that the intervention group, to which each participant belongs, was ignored in the analysis carried out and in the discussion of the results obtained.

Author response: Thank you for your comment. Our study aimed to explore the possible correlation between weather fluctuations and daily step counts over two years. We provided pedometers to only the intervention groups to track their daily steps, so we do not have a control group for comparison. Hence, we conducted an observational study using the cohort and did not intend to compare the intervention groups with each other.

Lines 81 to 88: Although a reference is included concerning a detailed description of the study, some brief explanation should be provided concerning the recruitment of participants. In the Introduction section authors consider individuals with prediabetes and type 2 diabetes, but in the Material and Methods there is no reference to those conditions.  Authors should also provide some explanation on why 385 participants were invited, but only 120 were included in the analysis. What was the inclusion/exclusion criteria?

Author response: Thank you for the suggestion. Based on your comment, we have added important details to the participant recruitment process. Lines 99-124

Line 89: Specify the intervention groups.

Author response: Thank you for your comment. We refer to the first comment response.

Line 108: Considering the study design, the intervention group should also be considered as an independent variable.

Author response: Thank you for your comment. We refer to the first comment response.

Line 123: The daily step count is the main outcome of this study, but only a descriptive at baseline is given. As the study took two years, I suggest the presentation of a more detailed analysis of the daily step count.

Author response: Thank you for your comment. We have included the details in the text of the method section; it is described in line with how steps were logged and monitored over two years. Lines 125-130

Lines 134 to 136: The intervention group should also be considered in Figure 1.

Author response: Thank you for your comment. We refer to the first comment response.

Lines 144 to 145: In table 2 I suggest the replacement of the T-value by the P-value.

Author response: We have included the P-value in the table based on your comment. Refer to table 2.

Lines 169 to 171: This statement does not seem to agree with the results presented in Table 2, and the conclusions presented in the first paragraph of the Discussion.

Author response: Thank you for the relevant comment. We have discussed why the disagreements occurred at the end of the first paragraph of the discussion; refer to lines 199-207.

  Line 181: The writing does not look right. Please verify.

Author response:  Thank you for the comment.

Lines 189 to 205: Since "The current study is a secondary analysis of the Sophia Step Study, a randomized controlled trial (RCT) aimed at promoting physical activity among people with prediabetes and type 2 diabetes", some discussion should be included regarding the impact of the interventions in daily steps during the two-year program. In evaluating the impact of weather conditions on the daily steps, the interventions that each participant was subjected to should be considered and discussed.

Author response: Thank you for your suggestion. We will clarify the text accordingly. Line 117-124

Lines 227 to 232: The conclusions presented in the Conclusions section are not in line with the conclusions presented in the first paragraph of the Discussion section. Please check the text.

Author response: Thank you for your suggestion. In the discussion, lines 190-195, we have clarified the text accordingly.

Round 2

Reviewer 1 Report

Comments and Suggestions for Authors

The changes made meet the requirements. The work can now be accepted after minor revisions requested by other reviewers

Author Response

Thank you for your valuable comments.

Reviewer 3 Report

Comments and Suggestions for Authors

In the revision version of the manuscript the authors addressed most of the comments made to the first version, however there are important comments that were not considered in the revised version.

The authors clarified that in this study only the two intervention groups were considered: a multi-component intervention group (self-monitoring of steps with counselling support), and a single-component intervention group (self-monitoring of steps without counselling support). Thus, the association between the intervention group and the number of steps (the outcome) should be taken into consideration. The assignment of participants to an intervention group (multi-component or single-component) cannot be ignored in the statistical analysis. The justification provided by the authors: “we conducted an observational study … and did not intend to compare the intervention groups with each other” is not adequate. First of all this is not an observational study: “The study was a three-armed parallel randomised RCT” as stated in reference Rossen et al. Int J Behav Nutr Phys Act (2021). The possible implications of considering data from a “parallel randomised RCT” should be considered in the statistical analysis and discussed in the Discussion section.

The other comment that was not answered, concerned the necessity of a more detailed descriptive analysis of the daily steps. The daily steps correspond to the main outcome of this study, but only a descriptive at the baseline was given. The authors should include a more detailed analysis, with at least the descriptive at the baseline and at the end of the study, for the two intervention groups.

Author Response

Response of the authors, second round

In the revision version of the manuscript the authors addressed most of the comments made to the first version, however there are important comments that were not considered in the revised version.

Authors response: We thank you for your confirmation.

Review comment: the authors clarified that in this study only the two intervention groups were considered: a multi-component intervention group (self-monitoring of steps with counselling support), and a single-component intervention group (self-monitoring of steps without counselling support). Thus, the association between the intervention group and the number of steps (the outcome) should be taken into consideration. The assignment of participants to an intervention group (multi-component or single-component) cannot be ignored in the statistical analysis. The justification provided by the authors: “we conducted an observational study … and did not intend to compare the intervention groups with each other” is not adequate. First of all this is not an observational study: “The study was a three-armed parallel randomised RCT” as stated in reference Rossen et al. Int J Behav Nutr Phys Act (2021). The possible implications of considering data from a “parallel randomised RCT” should be considered in the statistical analysis and discussed in the Discussion section.

Authors response: Again, thank you for showing genuine interest in our work. Based on your recommendation, we analyzed the Sophia Step Study (RCT) intervention groups and included them in our model to determine whether there were any differences in steps between them. We regret to inform you that the analysis did not reveal significant differences between the groups, and the results aligned with our previous study. Nonetheless, we have highlighted the changes we made in green in the manuscript for your review.

Review comment: The other comment that was not answered, concerned the necessity of a more detailed descriptive analysis of the daily steps. The daily steps correspond to the main outcome of this study, but only a descriptive at the baseline was given. The authors should include a more detailed analysis, with at least the descriptive at the baseline and at the end of the study, for the two intervention groups.

Authors response: Again, thank you for the comment. We have decided only to provide the average number of participants' daily steps. Unfortunately, we cannot choose a specific day as the baseline or the last day to display the steps taken before and after the intervention. This is because all participants monitored their steps daily from the beginning to the end of the intervention. Additionally, it is challenging to have such a description since all participants started the intervention at different times over seven rounds between 2013 and 2020. However, we have included a graph displaying the step pattern between the two intervention groups over two years, Figure 1.